# Tillage-Induced Fragmentation of Large Soil Macroaggregates Increases Nitrogen Leaching in a Subtropical Karst Region

Shuangshuang Xiao [1,2,†], Xiajiao Liu [1,3,†], Wei Zhang [1], Yingying Ye [4], Wurong Chen [2] and Kelin Wang [1,*]

1   Key Laboratory of Agro-Ecological Processes in Subtropical Region, Institute of Subtropical Agriculture, Chinese Academy of Sciences, Changsha 410125, China
2   Guangxi Key Laboratory of Earth Surface Processes and Intelligent Simulation, Nanning Normal University, Nanning 530001, China
3   University of Chinese Academy of Sciences, Beijing 100039, China
4   College of Resources and Environment, Hunan Agricultural University, Changsha 410125, China
*   Correspondence: kelin@isa.ac.cn
†   These authors contributed equally to this work.

**Abstract:** Tillage leads to rapid loss of soil nitrogen (N) over a short period of time in karst areas. N leaching is the primary pathway of soil N loss and therefore is key to understanding the mechanisms of N loss induced by tillage. However, the factors affecting N leaching under tillage are not fully understood. Effects of tillage at various frequencies on leached N were examined in a one-year in situ simulation experiment using five tillage treatments: no tillage (T0), semiannual tillage (T1), and tillage every four months (T2), two months (T3), and monthly (T4). Concentration and amount of leached N had peaks in dry–rewetting months. Tillage significantly increased total amounts of leached N during the one-year experiment, and the largest amount of leached N was under tillage at the highest frequency. The primary form of N in leachate was $NO_3^-$ (88.49–91.11%), followed by DON (7.80–9.87%), and then $NH_4^+$ with the lowest amount (1.09–2.10%). Tillage increased the amount of leached $NO_3^-$ and DON, but had no significant effect on leached $NH_4^+$. Additionally, the amount of leached N had significantly negative correlations with 5–8 mm soil aggregate, $NO_3^-$, DON, and sand content, and positive correlations with 2–5 and 0.25–2 mm. Soil 5–8 mm aggregate and DON were the main factors explaining the variation in leached N according to the RDA analysis. Tillage increased the breakdown of large aggregates, appearing to have increased the mineralization of organic matter, which resulted in increased N leaching. Our results emphasize the importance of reducing or eliminating physical disturbance indued by tillage and maintenance of large soil aggregates for decreasing N leachate in lime soil of karst regions.

**Keywords:** karst ecosystem; human disturbance; nitrate; leachate; aggregate

## 1. Introduction

Nitrogen (N) is one of the most common and important nutrient elements in terrestrial ecosystems, and its biogeochemical cycle is the focus of current research [1]. When agriculture replaces natural ecosystems, soil N decreases rapidly [2]. Leaching is as an important pathway of soil N loss that not only restricts the sustainable use of soil but also leads to eutrophication of groundwater [3]. Therefore, it is critical to understand the mechanisms of leaching losses of soil N during the initial stage of conversion to agriculture.

Tillage is a common agricultural practice that is closely associated with leaching of soil N [4,5]. However, the mechanisms of soil N leaching induced by tillage have not been fully revealed, and results of previous studies are inconsistent. Conventional tillage can promote soil N mineralization and increase N leaching [4], and reduced tillage can decease N leaching [6,7]. By contrast, others find that traditional plowing reduces soil N leaching and that less or no tillage increases N in leachate [5,8]. Such inconsistent effects of tillage on leaching of soil N may depend on regional climate, soil properties, and the method and

intensity of plowing [5]. Therefore, the effects of tillage on the leaching of soil N remain not fully understood.

Total N in leached solutions is generally called total dissolved N (TDN) and has three forms: nitrate ($NO_3^-$), ammonium ($NH_4^+$), and dissolved organic N (DON). Negatively charged $NO_3^-$ is not easily absorbed by soil particles but is highly soluble in water and is the primary form of N in leachate in drylands [9]. Positively charged $NH_4^+$ is easily adsorbed by colloids and fixed by mineral crystals in soil. Therefore, it is not as easily leached as $NO_3^-$. However, when the soil adsorption capacity of $NH_4^+$ reaches saturation, $NH_4^+$ is leached from soil through macropores [10]. Dissolved organic N can also be the primary form of N in leachate [11,12]. Forms of soil N in leachate vary greatly with differences in land use types, soil properties, and soil N status [8,13]. Therefore, the amounts and proportions of different forms of N ($NO_3^-$, $NH_4^+$, and DON) in leachate under tillage in different areas require further study.

Tillage often alters soil properties and thus affects N leaching [3]. Soil aggregates are the basic units of soil's physical structure and are directly affected by tillage [14]. However, few studies have examined the relationship between aggregates and N leaching. Aggregates of different sizes are closely associated with soil porosity [15], and soil porosity greatly affects water transport and N leaching [16]. Thus, soil aggregates may be an important factor affecting the leaching of N. Revealing the effects of soil aggregates on N leaching can provide supplementary information for related studies. Other soil properties also affect the leaching of N, including soil texture, dissolved N and so on. Texture affects N leaching mainly via water migration and infiltration in soil. Sandy soils tend to be at a greater risk of N leaching than cohesive soils [17]. The soil content of dissolved N (such as $NH_4^+$ and $NO_3^-$) may also be correlated with N leaching [13]. Therefore, key factors affecting leaching loss of soil N under tillage need to be thoroughly studied.

Karst areas cover about 12% of the Earth's surface, and the largest karst region is in Southwestern China [18]. Because of tremendous population pressure, excessive agricultural cultivation has caused rapid loss of soil organic matter, serious rocky desertification and other environmental problems [19]. Zhang et al. [20] found that once a natural karst ecosystem is cultivated, soil N declines rapidly over a short period of time. The mechanism for its rapid loss is still not fully understood. Leaching is the primary pathway of soil N loss. Therefore, revealing how N leaching is affected by tillage is the key to understanding the mechanisms of soil N loss. We hypothesized that the amount of N leaching would increase as the frequency of tillage increased (Hypothesis I). Since a previous study showed that increased soil disturbance improved the number of fine aggregates, which promoted N mineralization [21], it might increase N leaching concentration. Hence, we hypothesized that a reduction in large aggregates caused by plowing is the primary factor accounting for increases in N leaching (Hypothesis II). In the present study, amounts and forms of leached N and soil physicochemical properties under tillage at various frequencies were measured. The major objectives were to determine (1) how N leaching loss changes with increasing tillage disturbance in karst areas and (2) the main drivers of N leaching during plowing. The results will provide a scientific basis for maintaining sustainable use of agricultural soils and preventing groundwater eutrophication.

## 2. Materials and Methods

### 2.1. Experimental Site

This experiment was conducted at the Huanjiang Observation and Research Station for Karst Ecosystems (108°18′–108°19′ E, 24°43′–25°44′ N), Chinese Academy of Sciences (GAS), Guangxi Province, China. The experimental site is a typical karst peak-cluster depression area with a flat depression surrounded by mountain ranges on three sides and an outlet in the northeast [22]. The climate is subtropical monsoon with rainy (April to September) and dry (October to March) seasons. Mean annual temperature is 18.5 °C, and mean annual precipitation is 1380 mm. The soil is developed from dolomite and is a Calcaric Leptic Regosols (Siltic) (IUSS Working Group WRB 2015), characterized as

calcareous, shallow, and gravelly. Soil texture ranges from clay at the bottom of the slopes to clay loam at the top (25–50% silt and 30–60% clay) [23]. Shrubland and shrub-grassland have been the dominant vegetation types in the area since 1985, as a result of spontaneous recovery when residents left the area [24]. Dominant plant species in the region are Clerodendrum serratum, Alangium chinense (Lour.) Harms, and Senecio scandens Buch. -Ham. ex D. Don.

## 2.2. Experimental Design and Field Management

A shrubland area (7 m × 40 m) was selected at the foot of a northwest-facing hill slope (inclination of almost 0°) on the basis of the lack of human disturbance, high organic carbon content, and relatively thick soil. These characteristics are representative of an area where future cultivation is possible. Soil properties before the experimental treatment was applied are shown in Table 1. In January 2014, 20 plots (2 m × 2 m each) were set up at the experimental site in a completely randomized block design with four replications. To prevent soil water and nutrient movement between plots, polyvinyl chloride (PVC) boards surrounded each plot. Each PVC board was 2 m-long and 0.5 m-high, with 0.35 m inserted into the soil and 0.15 m left above the ground. To eliminate potential effects of plants and to investigate the exact role of aggregate structure disruption triggered by tillage in soil N leaching, all aboveground vegetation was removed. However, to stimulate a farming microenvironment, plastic corn plants were used to mimic the shade provided by the crowns and to prevent raindrop splash erosion. To simulate the real cover environment, artificial plants were placed in the field for double planting season according to the actual cropping time. The first crop was from February to May, and the second crop was from June to September. Two, four, and six leaves were placed in artificial plants at the jointing stage, bell stage, and milk stage, respectively. Starting in June 2014, all plots received the following treatments: no tillage (T0; control (CK)); semiannual tillage (T1); tillage every four months (T2); bimonthly tillage (T3), and monthly tillage (T4). Tillage was performed at a depth of 15 cm with a manual coulter, as it is widely used in this region. The tillage-simulating experiment was conducted for one year. Detailed information on the field area and experimental design can be found in previous studies [14,22]. Monthly precipitation was obtained from a weather station at the Huanjiang Observation and Research Station for Karst Ecosystems, CAS.

**Table 1.** Soil physical and chemical properties before the experiment was applied ($n = 20$).

|  | SOC | TN | $NH_4^+$ | $NO_3^-$ | TDN | E-Ca | pH | Aggregate Size (%) | | | |
|---|---|---|---|---|---|---|---|---|---|---|---|
|  | (g kg$^{-1}$) | (g kg$^{-1}$) | (mg kg$^{-1}$) | (mg kg$^{-1}$) | (mg kg$^{-1}$) | (g kg$^{-1}$) |  | 5–8 | 2–5 | 0.25–2 | <0.25 |
| Mean | 37.01 | 3.17 | 31.85 | 28.01 | 77.66 | 2.97 | 6.4 | 35.38 | 43.19 | 20.24 | 1.2 |
| SE | 0.38 | 0.02 | 0.85 | 1.17 | 0.93 | 0.05 | 0.02 | 0.38 | 0.33 | 0.19 | 0.04 |

SOC, TN, $NH_4^+$, $NO_3^-$, TDN, and E-Ca indicated soil organic carbon, total nitrogen, ammonium nitrogen, nitrate nitrogen, and exchangeable calcium in soil, respectively.

## 2.3. The Leachate Collection and Analyses

Leachate was collected using a steel frame (50 cm in length, 30 cm in width, and 5 cm in height) combined with a sealed bottle underneath (Figure 1). Frames were installed at 30 cm in depth, with one frame per subplot. These frames were filled with quartz sand, and pieces of mesh nylon fabric were tied at the bottom to ensure that the sand did not wash away. At the bottom of the plate, there was an outlet connected with a sealed bottle underneath through a plastic pipe (2 mm in diameter). The sealed bottle was 2 L, according to the recent maximum monthly rainfall, to ensure that the collected leachate water would not overflow. This set up was buried underground (to 70 cm), with two pipes (0.5 mm in diameter) within the sealed bottle extending vertically aboveground. One pipe was an inlet for air, and the other was used to pump seepage water from the bottle. Leachate samples, which were filtered by the quartz sand inside the frames, were collected in the sealed bottles and then pumped to the surface every month. Samples were

stored in the refrigerator at $-18\ ^{\circ}$C for cryopreservation prior to chemical analyses. Before the experiment, the leaching water was taken and the leaching content of nitrogen was determined as nitrate (68.62 mg kg$^{-1}$), ammonium (3.26 mg kg$^{-1}$), and dissolved organic nitrogen (7.28 mg kg$^{-1}$). The leachates were collected for ten months, with the exception of October and December 2014, when leachate collection was insufficient due to low rainfall. Therefore, November 2014 and January of the following year were dry–rewetting months in our study.

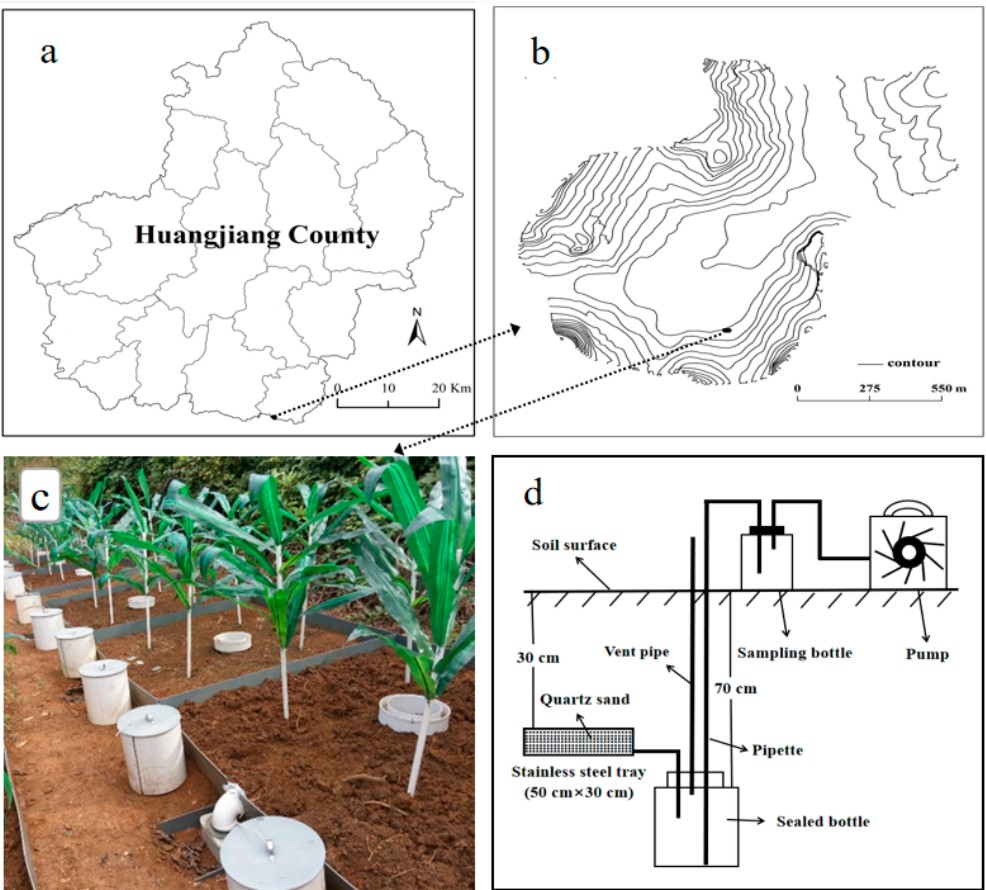

**Figure 1.** Map of region and field area. (**a**) Location map of the studied catchment within Huanjiang County. (**b**) Location of designed field. (**c**) Plot construction completed. (**d**) The soil-leachate-collecting device.

After the samples were thawed, TDN, NO$_3^-$, and NH$_4^+$ in leachate were determined by a continuous flow analyzer (AA3, SEAL, Norderstedt, Germany). The concentration of DON was calculated by subtracting inorganic N (NO$_3^-$ and NH$_4^+$) from TDN.

The following equation was used to calculate the total amount of N leached:

$$F = C \cdot V / S \tag{1}$$

where $F$ is the total amount of N leaching (mg m$^{-2}$); $C$ is the concentration of N leaching (mg L$^{-1}$); $V$ is the amount of leaching water (L); $S$ is the leachate collection area (0.15 m$^2$).

### 2.4. Soil Sampling and Analyses

All soil samples were collected from the top 0–10 cm soil layer in June 2015. Undisturbed soil samples were collected for aggregate analysis using a flat spade and stored in crush-resistant airtight containers. In the laboratory, visible roots, organic residues, and stone fragments in soil samples were removed manually. Soil samples were then gently crushed by hand along their natural breakpoints and passed through an 8 mm sieve

to determine the aggregate fractionation. Soil aggregates separated by dry sieving may more closely resemble those from fields in the absence of irrigation (e.g., our simulated corn field) [25]. Thus, we chose to perform soil aggregate separation using a dry-sieving procedure (Wang et al., 2015). Soil samples (approximately 500 g) were placed on a nest of sieves (5, 2, and 0.25 mm) on a dry-sieving instrument called Sieve Shaker (ST-A100, Xuxin, China). The sieves were mechanically shaken (amplitude = 1.5 mm) for 2 min. Aggregates retained on each sieve were collected and thus separated according to aggregate size (5–8, 2–5, 0.25–2, and <0.25 mm).

Composite soil samples were collected by mixing five soil samples in each plot. Soils were passed through a 2 mm mesh sieve, and then were divided into two portions for further processing. One portion was air-dried and passed through a 0.15 mm sieve for measurements of soil organic carbon (SOC), total N (TN), and soil pH. SOC was measured by wet oxidation with the dichromate redox colorimetric method [26]. Soil TN was determined with an Element Auto-Analyser (Vario MAX CN; Elementar, Hanau, Germany). Soil pH was measured with a pH meter (FE20K, Mettler-Toledo, Greifensee, Switzerland). The other portion was stored at 4 °C for analyses of $NH_4^+$, $NO_3^-$, and TDN. $NH_4^+$, $NO_3^-$, and TDN were extracted from the soil samples using 2 M KCl and measured by a continuous flow Auto Analyzer (AA3, SEAL, Norderstedt, Germany). Soil DON was equal to the difference between TDN and total inorganic N ($NH_4^+$ and $NO_3^-$). Exchangeable calcium (Ca) was displaced via compulsive exchange in 1 mol $L^{-1}$ ammonium acetate at pH 7.0 and analyzed by inductively coupled plasma atomic emission spectroscopy [27]. Soil texture was determined by using a laser diffraction particle size analyzer (Mastersizer, 2000, Malvern, UK).

### 2.5. Statistical Analyses

All statistical analyses were performed using SPSS 16.0 (SPSS Inc., Chicago, IL, USA). Data were logarithmically transformed before analysis, as needed, to improve normality and homogeneity of variance. Changes in soil aggregates, monthly amounts of TDN leaching, and annual amounts and composition of TDN leaching were compared among tillage treatments using one-way ANOVA. Fisher's least significant difference (LSD) post hoc test was used to identify significant differences among various treatments at a 5% level of significance. Pearson correlation was used to analyze the relationship between soil properties and the amounts of leached nitrogen. To gain insight into the impact of soil properties on nitrogen in leachate (including $NO_3^-$, DON, and $NH_4^+$), redundancy analysis (RDA) was performed in Canoco 5.0 (Centre for Biometry, Wageningen, The Netherlands). The soil property parameters that had no significant correlation with leached nitrogen were removed. The most discriminating explanatory variables were selected by the conditional effects test. Statistical significance of the RDA results was determined using Monte Carlo permutation, based on 999 runs with randomized data.

### 3. Results

#### 3.1. Effects of Tillage Disturbance on Soil Properties

After one year of the simulation experiment, the percentage of soil 5–8 mm aggregate decreased, and the percentage of 0.25–2 mm aggregate increased with the increase in tillage frequency (Table 2). Soil TN, $NO_3^-$, and DON were the highest under T0, followed by T1, T2, and T3, and T4 was the lowest. SOC and the rate of sand decreased with increasing tillage frequency. By contrast, the soil $NH_4^+$, exchangeable Ca, and pH showed no significant differences.

**Table 2.** Soil properties under different tillage frequencies after one-year simulation experiment. T0, no tillage; T1, semiannual tillage; T2, tillage every four months; T3, bimonthly tillage; T4, monthly tillage. Different letters indicate significant differences among treatments ($p < 0.05$). Values in parentheses are standard errors ($n = 4$).

| | T0 | T1 | T2 | T3 | T4 |
|---|---|---|---|---|---|
| Aggregate 5–8 (%) | 38.84 (3.66) a | 26.40 (2.05) bc | 27.76 (2.12) b | 25.16 (0.65) bc | 20.55 (1.43) c |
| Aggregate 2–5 (%) | 36.48 (1.25) b | 43.47 (0.87) a | 42.17 (0.82) a | 41.57 (1.37) a | 43.03 (0.82) a |
| Aggregate 0.25–2 (%) | 22.00 (2.58) c | 28.23 (1.56) b | 28.75 (2.02) ab | 31.22 (0.69) ab | 34.17 (1.22) a |
| Aggregate < 0.25 (%) | 2.69 (0.29) a | 1.90 (0.17) ab | 1.74 (0.21) b | 2.05 (0.48) ab | 2.25 (0.06) ab |
| TN (g kg$^{-1}$) | 3.02 (0.14) a | 2.70 (0.13) ab | 2.74 (0.15) ab | 2.57 (0.10) b | 2.50 (0.02) b |
| NO$_3^-$ (mg kg$^{-1}$) | 39.99 (3.86) a | 24.04 (3.58) bc | 28.67 (2.67) b | 16.06 (1.46) cd | 13.25 (0.82) d |
| DON (mg kg$^{-1}$) | 11.90 (1.49) a | 7.31 (0.58) b | 12.08 (0.44) a | 8.02 (1.93) b | 7.42 (1.24) b |
| NH$_4^+$ (mg kg$^{-1}$) | 2.45 (0.39) a | 4.52 (1.26) a | 2.91 (0.98) a | 2.59 (0.64) a | 2.77 (0.85) a |
| SOC (g kg$^{-1}$) | 36.10 (2.20) a | 29.93 (2.70) ab | 30.38 (2.88) ab | 27.32 (1.71) b | 27.53 (0.88) b |
| E-Ca (g kg$^{-1}$) | 2.73 (0.23) a | 2.61 (0.38) a | 2.64 (0.46) a | 2.34 (0.16) a | 2.32 (0.21) a |
| Sand (%) | 13.86 (1.27) a | 10.33 (0.98) ab | 10.17 (1.07) b | 10.10 (1.56) b | 8.86 (0.95) b |
| Soil pH | 6.51 (0.16) a | 6.46 (0.13) a | 6.39 (0.11) a | 6.43 (0.14) a | 6.44 (0.13) a |

E-Ca is exchangeable calcium in soil.

### 3.2. Dynamic Changes in Nitrogen Leaching under Tillage

The concentration of TDN in leachate was the highest in January (37.23–74.02 mg L$^{-1}$), followed by March (23.17–61.03 mg L$^{-1}$) and November (33.05–47.31 mg L$^{-1}$) (Figure 2). The amount of TDN in leachate in all treatments showed several peaks. The maximum peak appeared in January 2015 (170.27–501.47 mg m$^{-2}$) when it was rewetting, and TDN amounts were notably higher under tillage treatments than no tillage. Small peaks in amounts of TDN in leachate occurred in November 2014 and May 2015. Tillage significantly affected amounts of N leaching in November 2014 and January and May 2015 ($p < 0.05$) by one-way ANOVA. In addition, the amount of TDN leaching increased significantly with the increase in water leaching ($R^2 = 0.40$; $p < 0.01$) in the dry season (October to March of the following year). However, there was no significant relationship between the amount of TDN leaching and volume of leached water in the rainy season (April to September).

### 3.3. Total Amount and Composition of Nitrogen in Leachate

Compared with T0, total amounts of TDN leaching under all tillage treatments increased significantly ($p < 0.05$) within the one-year experiment (Figure 3). The highest total amount of TDN leaching was 2852 mg m$^{-2}$ in T4, which was 1.21 times greater than that in T0. The amounts of TDN leaching under T1, T2, and T3 were 2114 mg m$^{-2}$, 1785 mg m$^{-2}$, and 1893 mg m$^{-2}$, respectively. Among the forms of N in leachate, NO$_3^-$ content was the highest in the five treatments at 88.5% to 91.1%. Dissolved organic nitrogen had the second highest content, accounting for 7.8% to 9.9%, and NH$_4^+$ had the lowest content at only 1.1% to 2.1%.

### 3.4. Soil Aggregate Fragmentation and Other Properties Affecting N Leaching

Pearson correlation showed that amounts of TDN, NO$_3^-$, and DON in leachate were significantly negatively related with 5–8 mm aggregates, soil NO$_3^-$, DON, and sand, and significantly positively related with 2–5 and 0.25–2 mm aggregates (Figure 4). On the contrary, there was no significant correlation between the amount of NH$_4^+$ leaching and any of the soil physicochemical properties except the soil DON.

The RDA identified soil 5–8 mm aggregate and DON as the major explanatory variables for the composition of N leaching ($p < 0.05$). All selected parameters together explained 62.54% of the variance, with axes 1 and 2 explaining 45.59% and 16.95% of the variance, respectively (Figure 5).

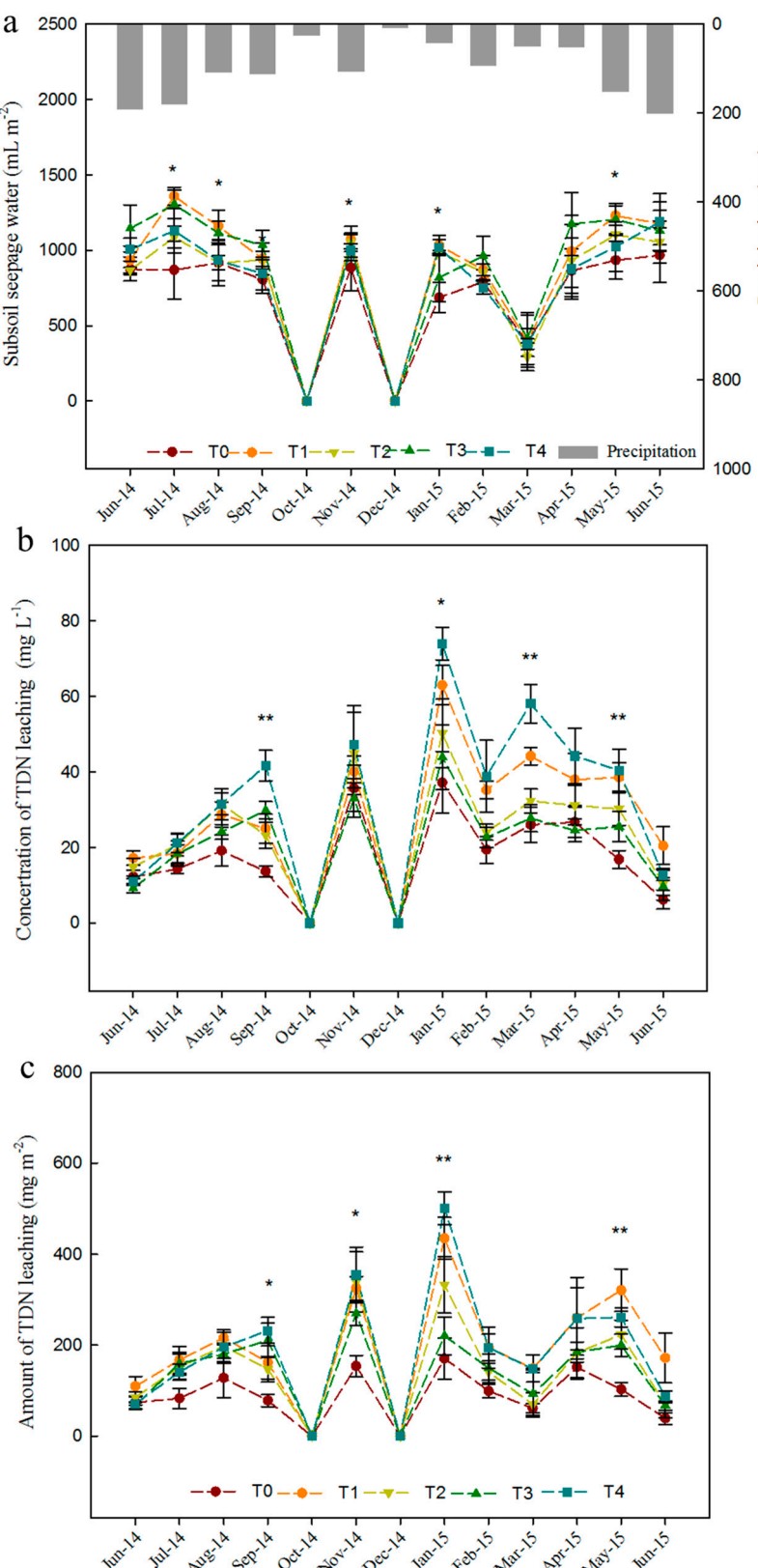

**Figure 2.** Precipitation and subsoil seepage water (**a**), concentration (**b**), and amount (**c**) of TDN leaching every month. T0, no tillage; T1, semiannual tillage; T2, tillage every four months; T3, bimonthly tillage; T4, monthly tillage. The asterisk (*) and (**) represent significant differences among different tillage treatments at the same time period (one-way ANOVA, $p < 0.05$ and $p < 0.01$, respectively). Bars represent mean ± standard error ($n = 4$). TDN represents total dissolved nitrogen.

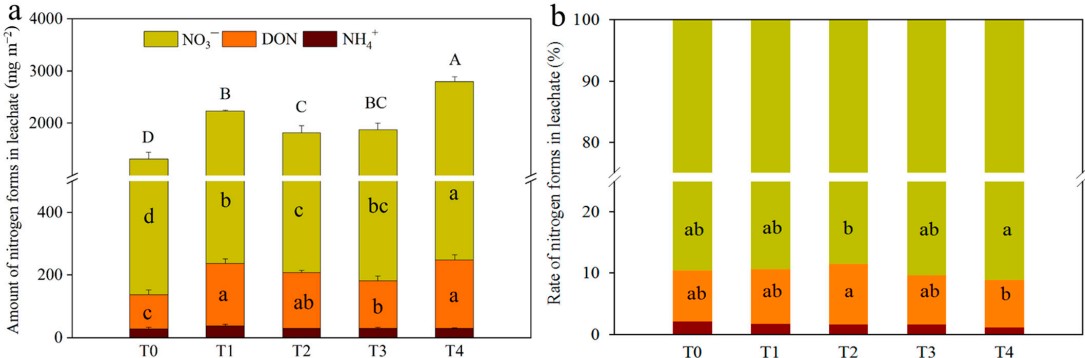

**Figure 3.** Amount (**a**) and rate (**b**) of nitrogen forms (including $NO_3^-$, DON, and $NH_4^+$) in leachate under tillage disturbance after 1 year. T0, no tillage; T1, semiannual tillage; T2, tillage every four months; T3, bimonthly tillage; T4, monthly tillage. Different letters indicate significant differences among treatments (one-way ANOVA, $p < 0.05$). Bars represent mean $\pm$ standard error ($n = 4$).

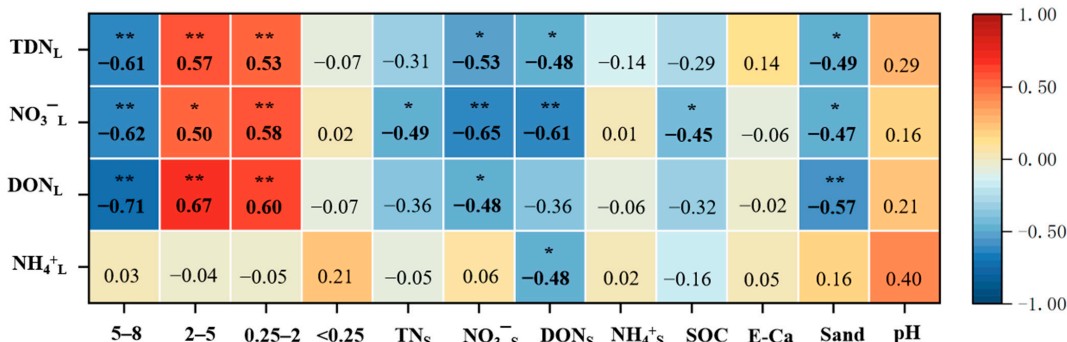

**Figure 4.** Correlation analysis of N leaching and soil physicochemical properties. The asterisk (*) and (**) represent that the correlation is significant ($p < 0.05$ and $p < 0.01$, respectively). $TN_S$, $NO_3^-_S$, $DON_S$, and $NH_4^+$ represent total nitrogen, nitrate, dissolved organic nitrogen, and ammonium in the soil after one-year simulation experiment. $TDN_L$, $NO_3^-_L$, $DON_L$, and $NH_4^+_L$ represent the accumulative amounts of total nitrogen, nitrate, dissolved organic nitrogen, and ammonium in leaching water during the one-year experiment. E-Ca represents exchangeable calcium in soil.

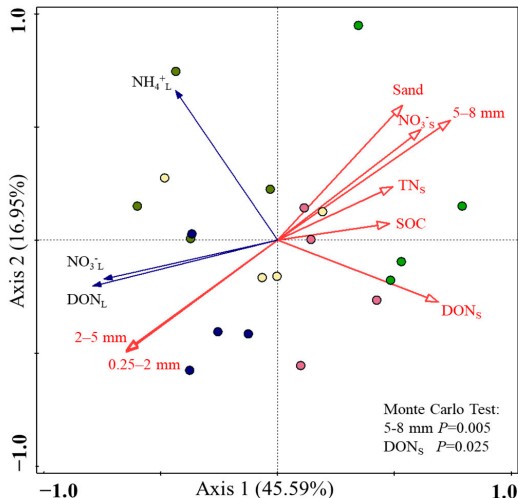

**Figure 5.** Effect of soil properties on nitrogen in leachate (including $NO_3^-$, DON, and $NH_4^+$) according to RDA analysis. $TN_S$, $NO_3^-_S$, and $DON_S$ represent total nitrogen, nitrate, and dissolved organic nitrogen in soil under tillage disturbance after one year. $NO_3^-_L$, $DON_L$, and $NH_4^+_L$ represent the cumulative amounts of nitrate, dissolved organic nitrogen, and ammonium in leaching water during the one-year experiment.

## 4. Discussion

### 4.1. Dynamic Characteristics of Soil N Leaching

In this study, the concentration and amount of TDN in leachate presented high levels in November and the following January (Figure 2). Those two months were post-drought, and soil experienced drying and rewetting. This result indicates that N leaching in a karst calcareous soil increased sharply when soils were rewetted following drought conditions. Leitner et al. [28] obtained a similar result in their study on N leaching in karst temperate forests. Previous studies had shown that rewetting of dry soils usually resulted in a significant loss of soil nutrients [29,30]. Gao et al. [31] pointed out that soil rewetting stimulates mineralization of soil organic nitrogen and the net nitrogen mineralization rate through meta-analysis. It greatly promoted the production of soluble nitrogen and led to an increase in nitrogen leaching. Additionally, our study found that tillage markedly increased the N amounts in leachate under dry–rewetting months. Tillage can destroy large aggregates, release fresh organic matter, and promote organic N mineralization, leading to rising soil dissolved N [14]. More soluble nitrogen could be accumulated under tillage with higher frequency during drought and be leached from the soil profile after dry soil is rewetted by sufficient rainfall. In this trial, seasonal variation in N leaching under tillage was largely caused by the uneven distribution of seasonal rainfall. With the increase in extreme weather events (e.g., drought and torrential rainfall) in recent decades, further research on the dynamics of N leaching under tillage is needed to maintain sustainable agriculture. Moreover, in actual agricultural production, N leaching may also be affected by fertilization level, sowing date, crop N uptake, growth season, etc. [32], but those factors were not involved in this study.

### 4.2. Annual Amount and Composition of Nitrogen Leaching under Tillage

We found that annual amounts of TDN leaching under tillage treatments were notably higher than under no tillage. The largest TDN leaching was under tillage at the highest frequency (T4). This is basically in line with our first hypothesis. Tillage can lead to continuous soil macropores, thereby increasing the number of water paths in soil [33]. As a result, downward water migration can transport more soil dissolved N. For example, Zhang et al. [13] observed a lower $NO_3^-$ content in surface soil (0–10 cm) under traditional tillage than under no tillage, whereas $NO_3^-$ content was higher in deep soil (10–60 cm) under traditional tillage than under no tillage. Those results indicate tillage can increase water flow paths, leading to the transport of soluble N to deeper soil or out of the soil. In addition, plowing can also increase soil aeration and promote nitrification of soil N [21]. As a consequence, more soil $NO_3^-$ is produced, which can dissolve in soil water and easily migrate, and be lost from the soil. However, another study found that increased tillage intensity did not lead to a rise in N leaching [5]. Trolove et al. [5] found traditional tillage can reduce $NO_3^-$ leaching by 40% compared with drilling tillage. They proposed that the soil under traditional cultivation was more susceptible to compaction by treading than that under drilling tillage. The compaction reduced $NO_3^-$ leaching because of an increase in N denitrification and production of $N_2O$ and $N_2$ gases [34].

In this study, $NO_3^-$ was the primary form of N in leachate in all treatments in the karst area, followed by DON and $NH_4^+$ with the lowest content. Nitrate is negatively charged, and therefore, absorption by soil particles is difficult, and it is readily soluble in water [9]. In addition, high soil pH in karst areas favors activity of nitrifying bacteria, which increases production of $NO_3^-$ [35]. Lime soil in karst regions also shows relatively low immobilization of $NO_3^-$ to recalcitrant organic N [36]. This allows $NO_3^-$ to accumulate easily in soil, and it is then lost in leachate with sufficient rainfall. Hence, $NO_3^-$ was the primary N form in leachate under all treatments. By contrast, the level of $NH_4^+$ leaching was very low. Because $NH_4^+$ is easily adsorbed by colloid and fixed by mineral cavitation in soil, it is not easily leached. Additionally, high soil pH in karst regions can promote nitrification of $NH_4^+$ and limits its accumulation, also contributing to low leaching levels [36]. In addition, we also found amounts of $NO_3^-$ and DON leaching

were significantly higher under tillage treatments than under no tillage, but there was no significant difference in $NH_4^+$. Our results show that tillage produced a decrease in larger aggregates and an increase in smaller aggregates along with an increase in N leaching. This suggests that manual tillage can promote the release and mineralization of N that was previously physically protected inside aggregates [21] and subsequently leached. Although plowing can also result in the mineralization of organic N to produce more $NH_4^+$, the high pH of lime soil in karst regions stimulates the oxidation of $NH_4^+$ to $NO_3^-$ [36]. Soil $NH_4^+$ content can reach a dynamic equilibrium and be maintained at a stable level, as has been previously observed [14]. Therefore, there was little difference in $NH_4^+$ in leachate under all treatments.

*4.3. Effects of Soil Properties on Nitrogen Leaching*

Our experiment results show that 5–8 mm soil aggregate (%) had a significantly negative correlation with leached nitrogen ($p < 0.01$). Additionally, 5–8 mm aggregate was the largest factor explaining the variation in soil nitrogen leaching according to RDA analysis. This is consistent with our second hypothesis. Fresh organic matter is exposed to air after fragmentation of large aggregates [37]. Without the physical protection provided by aggregates, organic matter (including organic N) is easily decomposed into simple organic (e.g., DON) and inorganic ($NO_3^-$, $NH_4^+$) compounds by microorganisms [21]. Ultimately, simple compounds leach down with moisture and increase soluble N content in leachate. Disruption of 5–8 mm large aggregates can also reduce the number of capillary pores [38], and therefore, the water-holding capacity of soil decreases, which increases water flow by gravity and transport of dissolved N. In addition, in this study, 2–5 mm and 0.25–2 mm soil aggregates had a significantly positive correlation with leached nitrogen. Plowing produces many small aggregates and decreases soil bulk density, which increase the number of macropores between soil aggregates [39]. As a result, it raises the paths of gravity of water migration and accelerates the loss of soluble N.

Soil DON was the second main factor explaining the variation in nitrogen leaching according to the RDA analysis. Soil DON and $NO_3^-$ had a significantly negative correlation with leached nitrogen ($p < 0.05$). Soil DON is one of the main substrates for nitrogen mineralization and nitrification, which affects the production of soil inorganic nitrogen. Additionally, DON and $NO_3^-$, as the main sources of leached nitrogen in soil, were accumulated in the topsoil after tillage disturbance. In the subtropical monsoon region (experimental area) with abundant rainfall, dissolved soil N (except $NH_4^+$) can easily migrate downward to the subsoil or even leach out of the soil layer. Zhang et al. [13] found higher amounts of N in leachate collected in the subsoil were associated with less soluble N in the upper soil, which is consistent with our result. In addition, there was a significant negative correlation between sand content and nitrogen leaching in our study, which is contrary to the result of Beaudoin et al. [40]. They found that higher soil sand content was associated with greater N leaching. The saturated water conductivity of sandy soil is high, and as more water passes through a unit area in a unit time, more dissolved N can be transported. However, tillage treatments (T2, T3, and T4) with lower sand content led to a greater annual volume of water leaching compared with CK (T0) in our study (Figure S1). We speculated that it was the crushing of large particles in the soil texture caused by tillage [41] that resulted in the loss of granular organic N and increased amounts of N leaching.

There was no significant relationship between exchangeable Ca in soil and N in leachate in our study. However, interestingly, there was a significant correlation between the content of Ca in leachate and the amount of leached nitrogen (Figure S2). In a previous study, it was found that $Ca^{2+}$ in the leaching solution was significantly correlated with the loss of organic carbon [22]. Organic carbon and nitrogen are the main components of organic matter. In karst areas, the soil is rich in carbonate (such as calcium carbonate), and its reprecipitation can reduce the utilization of organic matter in the aggregates by microorganisms and promote the accumulation of soil organic matter [42]. However, the

mineralization of soil organic matter under tillage resulted in the increase in $CO_2$, which promoted the dissolution of carbonate and the leaching of $Ca^{2+}$. This eventually resulted in a significant correlation between $Ca^{2+}$ in leached solution and leached N.

## 5. Conclusions

The concentration and amount of leached nitrogen in soil had peaks in dry–rewetting months. Additionally, tillage remarkably increased the amount of TDN leaching in rewetting months. Total amount of TDN leaching in one year under tillage treatments were markedly higher than that under no tillage, and the greatest amount of leached N occurred with the highest frequency of plowing (T4). The compositions of N leaching were mainly $NO_3^-$, followed by DON and $NH_4^+$ with the lowest content. Contents of $NO_3^-$ and DON in leachate were notably higher under tillage than those under no tillage, whereas tillage did not significantly affect $NH_4^+$ leaching. There were significantly negative correlations between the amount of leached N and the contents of 5–8 mm soil aggregate, $NO_3^-$, DON, and sand. It had significantly positive correlations between the amount of leached N and 2–5 mm and 0.25–2 mm soil aggregates. The 5–8 mm aggregate and DON were the main factors explaining the variation in leached nitrogen according to RDA analysis. Our results support both hypotheses tested: the amount of leaching increased as frequency of tillage increased (supporting Hypothesis 1), and tillage increased the breakdown of large aggregates, which appears to have increased the mineralization of organic matter, which resulted in increased N leaching (supporting Hypothesis 2). The findings suggest that leached N should receive more attention in dry–rewetting months, and conservation tillage or no tillage should be emphasized as important management methods to reduce soil N leaching.

**Supplementary Materials:** The following supporting information can be downloaded at: https://www.mdpi.com/article/10.3390/land11101648/s1. Figure S1: Volumes of leached water during one-year experiment; Figure S2: Relationship between the amounts of $Ca^{2+}$ leaching and TDN leaching according to Pearson correlation analysis.

**Author Contributions:** Conceptualization, S.X. and X.L.; methodology, S.X. and X.L.; software, Y.Y.; investigation, W.C.; resources, S.X. and K.W.; data curation, X.L.; writing—original draft preparation, S.X. and X.L.; writing—review and editing, S.X. and W.Z.; supervision, K.W. All authors have read and agreed to the published version of the manuscript.

**Funding:** This research was supported by the National Natural Science Foundation of China (U20A2011; 41930652; 41807517), Special Fund Projects of Central Government Guiding Local Science and Technology Development (Guike ZY20198012), and the Guangxi Natural Science Foundation (2020GXNSFDA238012; 2018GXNSFBA281157).

**Institutional Review Board Statement:** Not applicable.

**Informed Consent Statement:** Not applicable.

**Data Availability Statement:** The data that support the findings of this study are available from the authors upon reasonable request.

**Conflicts of Interest:** The authors declare no conflict of interest.

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
