# Peer review of "Tillage-Induced Fragmentation of Large Soil Macroaggregates Increases Nitrogen Leaching in a Subtropical Karst Region"

_land, doi:10.3390/land11101648_

Round 1
Reviewer 1 Report
Reviewers Comments on Xiao et al article for land
There is some nice clear results in this experiment. However I feel the main messages are a bit hidden in the detail that the authors provide in trying to explain all of their results. I have suggested some sentences that should be added to the abstract, discussion and conclusion to highlight the key findings of this paper.
Once these are added, and the calculations double-checked (the N leaching rates are incredibly high), then I think that the paper is okay to be published. If no mistake is found in the calculations, then some explanation must be provided for the incredibly high N leaching rates.
The English is understandable, but an edit by someone with excellent English would help the readability of this paper.
Abstract
After the words “… by RDA analysis.” I would add the following sentence. Tillage increased the breakdown of large aggregates, which appears to have increased the mineralisation of organic matter, which resulted in increased N leaching.”
Background
No mechanistic explanation is given in the background as to how a “Reduction of large aggregates caused by plowing would be the primary factor accounting for increases in N leaching (Hypothesis II).” Nitrogen leaching is the product of drainage x concentration of TDN. The authors need to clearly explain how a reduction in large aggregates might increase either drainage or TDN concentration. A possibility is that the increased soil disturbance increased the amount of fine aggregates, which increased N mineralisation, see Curtin et al. 2014 (#33 cited in your reference list).
Methodology
Looks fine. I loved the use of plastic plants to simulate plant cover, while avoiding any confounding issues with N uptake.
Results
Calculations and units must be checked, are they mg or g per m2? The unit used in this paper is g m-2 (except line 226), which gives incredibly high leaching results of over 21,000 kg TDN/ha. However, 21 kg TDN/ha would seem much too low for leaching over a 1 year fallow.
Discussion
Section 4.2 1st paragraph. You need to mention that your results showed that tillage increased drainage [which leads to increased N leaching provided the N concentration remains unchanged].
I would rewrite lines 313-315 as “Our results showed that tillage produced a decrease in larger aggregates and an increase in smaller aggregates along with an increase in N leaching. This suggests that manual tillage can promote the release and mineralization of N that was previously physically protected inside aggregates [33], which was subsequently leached.
Section 4.3 Final paragraph
In the opening sentence, leaching of exchangeable Ca is dismissed, since there was no significant difference among the treatments. However, the mean exchangeable Ca concentration still dropped by a massive 0.4g/kg, which accounts for approximately 1,200 kg Ca/ha! This may be important, depending on the accuracy of calculations of the extremely high N losses.
Conclusion
For clarity, the conclusion should state whether your results proved or disproved your two hypotheses under test.
e.g. our results supported both hypotheses under test
the amount of leaching increased as frequency of tillage increased (supporting hypothesis 1)
Tillage increased the breakdown of large aggregates, which appears to have increased the mineralisation of organic matter, which resulted in increased N leaching. (supporting hypothesis 2)
Suggested improvement to the Discussion (up to the authors and Editors if they think this is essential).
In my opinion, the key message of the research could be clearly explained in the Discussion, (probably included in section 4.2) if structured as follows.
N leaching is the product of drainage x total dissolved N concentration
Tillage increased drainage [perhaps due to a decrease in soil water holding capacity, or an increase in bypass (macropore) flow or other possibilities?] The increase in drainage explained XX% of the increase in leaching. [Calculate XX% by assuming the increased drainage had the same N concentration as the no tillage treatment]
Tillage also resulted in an increase in N concentration. This is believed to be the result of increased mineralisation of organic matter resulting from the breakdown of large aggregates. This explained the remaining YY% of the increase in leaching.

Reviewer 2 Report
The work „Effects of physical disturbance caused by tillage on soil nitrogen leaching in a subtropical karst region“ is interesting and concerns an important ecological aspect, namely – washing out the nitrogen from soil.
A main shortcoming of this experiment is the fact that research is being carried out for only one year, which decreases the credibility and repetition of results.
Another drawback is the lack of plants, which means that the experiment is unrelated to natural conditions. In nature, there are only short periods of time in which plants aren’t present and only plants (living organisms) are retaining nitrates (biological sorption).
In my mind, authors relying on the results of an yearly experiment are drawing too far fetched conclusions. I suggest using initial studies on nitrogen wash out.
After an analysis of the experiment, I have the following comments, that are required to pass to the next stage of procedure:
· Lack of analysis of physical and chemical soil samples, taken from testing grounds before the experiment – to complete
· Give the composition of the soil solution before setting up the experiment
· The soil developed from dolomite and is a Rendzina (IUSS Working Group WRB 2015) – this is not a WRB classification – please use the correct, full WRB classification
· Lack of detailed weather conditions – temperature.
Reviewer 3 Report
A very valuable paper! But again my standard comment referring to a quotation of Carl Friedrich Gauss (1777-1855) ... "Nothing shows mathematical incomprehension more clearly than excessive accuracy in numerical calculation."
PLEASE limit the use of decimals throughout the text and in all tables and all figures as follows:
123
12.3
1.23
0.123
Reviewer 4 Report
The manuscript is generally well written and easy to follow. I have some minor suggestions that the authors may reconsider.
1. The TITLE is not clear. The "physical disturbance" is actually presented as soil properties in the main text.
2. Table 1, the top four lines, 5-8, 2-5, 0.25-2, <0.25, are not clear. Please add soil aggregates.
3. L21 and in the main text, the define of "dry-rewetting months" is not clear.
Round 2
Reviewer 2 Report
The authors answered all my questions except for the WRB classification.
Please provide the complete WRB soil classification with principal qualifiers and supplementary qualifiers. After applying these corrections, I accept the paper for printing.
